# The Impact of Iron Supplementation for Treating Anemia in Patients with Chronic Kidney Disease: Results from Pairwise and Network Meta-Analyses of Randomized Controlled Trials

**DOI:** 10.3390/ph13050085

**Published:** 2020-04-30

**Authors:** Marcel Adler, Francisco Herrera-Gómez, Débora Martín-García, Marie Gavid, F. Javier Álvarez, Carlos Ochoa-Sangrador

**Affiliations:** 1Center for Medical Oncology & Hematology, Hospital Thun, 3600 Thun, Switzerland; marcel.adler@spitalstsag.ch; 2Pharmacological Big Data Laboratory, University of Valladolid, 47005 Valladolid, Spain; marie.gavid@chu-st-etienne.fr (M.G.); alvarez@med.uva.es (F.J.Á.); 3Nephrology, Complejo Asistencial de Zamora, 49022 Zamora, Spain; 4Nephrology, University Clinical Hospital of Valladolid, 47005 Valladolid, Spain; deboramarg@yahoo.es; 5Anatomy, Faculty of Medicine Jacques Lisfranc, Jean Monnet University, 42270 Saint-Priest en Jarez, France; 6Ethics Committee for Drug Research - East Valladolid, University Clinical Hospital of Valladolid, 47005 Valladolid, Spain; 7Clinical Epidemiology Research Support Office, Complejo Asistencial de Zamora, 49022 Zamora, Spain; cochoas2@gmail.com

**Keywords:** anemia, iron-deficiency, iron compounds, Kidney Diseases

## Abstract

After relative erythropoietin deficiency, iron deficiency is the second most important contributing factor for anemia in chronic kidney disease (CKD) patients. Iron supplementation is a crucial part of the treatment of anemia in CKD patients, and intravenous (IV) iron supplementation is considered to be superior to per os (PO) iron supplementation. The differences between the available formulations are poorly characterized. This report presents results from pairwise and network meta-analyses carried out after a comprehensive search in sources of published and unpublished studies, according to the Preferred Reporting Items for Systematic reviews and Meta-Analyses (PRISMA) recommendations (International prospective register of systematic reviews PROSPERO reference ID: CRD42020148155). Meta-analytic calculations were performed for the outcome of non-response to iron supplementation (i.e., hemoglobin (Hgb) increase of <0.5–1.0 g/dL, or initiation/intensification of erythropoiesis-stimulating agent (ESA) therapy, or increase/change of iron supplement, or requirements of blood transfusion). A total of 34 randomized controlled trials (RCT) were identified, providing numerical data for analyses covering 93.7% (n = 10.097) of the total study population. At the network level, iron supplementation seems to have a more protective effect against the outcome of non-response before the start of dialysis than once dialysis is initiated, and some preparations seem to be more potent (e.g., ferumoxytol, ferric carboxymaltose), compared to the rest of iron supplements assessed (surface under the cumulative ranking area (SUCRA) > 0.8). This study provides parameters for adequately following-up patients requiring iron supplementation, by presenting the most performing preparations, and, indirectly, by making it possible to identify good responders among all patients treated with these medicines.

## 1. Introduction

After relative erythropoietin deficiency, iron deficiency is the second most common contributing factor for anemia in chronic kidney disease (CKD) patients [1,2]. Anemia may be considered a surrogate marker of CKD severity, and its treatment may slow down the progression of concomitant heart disease and cardiovascular disease, as well as the evolution of CKD towards end-stage kidney disease (ESKD) [3].

Intravenous (IV) preparations are the preferred galenic formulation for iron supplementation both before and once dialysis is initiated [4]. However, the effects and benefits of different IV iron supplements compared to per os (PO) iron supplements are still poorly characterized in the different stages of CKD: an accurate description of the impact of different iron supplementation formulations may improve physicians’ decision-making process and promote an individualized treatment approach to iron-deficiency anemia (IDA) in CKD patients.

Our study aim was to assess treatment response to different commercially available iron supplements in CKD patients, which was defined by the increase in the hemoglobin (Hgb) level and/or the need for erythropoiesis-stimulating agent (ESA) therapy and other treatments of anemia in CKD patients.

## 2. Results

The standardized flowchart produced by the Preferred Reporting Items for Systematic reviews and Meta-Analyses (PRISMA) group [5] is presented in Figure 1, showing the study selection process that led to the inclusion of 34 randomized controlled trials (RCT). All studies investigating iron supplementation in patients with CKD were screened for eligibility. Non-relevant articles (e.g., non-research studies, observational studies), as well as studies with different investigated outcomes, were excluded. No unpublished studies were found (i.e., meeting abstracts, PhD and Master Theses, and industry reports did not provide other studies different than that published). However, in some cases more than one published article/unpublished report presented the results from one study, of which extension follow-up studies/post-hoc analyses were identified for three studies: ferumoxytol authorization studies [6,7,8,9,10], Dialysis patients’ Response to IV iron with Elevated ferritin (DRIVE) and DRIVE II studies [11,12], and King’s College Hospital (KCH)/Royal Adelaide Hospital (RAH) studies [13,14,15]. Study participants details and the characteristics of the eligible studies, as well as details of the interventions, comparators and all outcomes evaluated in the included trials, as emanate from our systematic narrative synthesis are available for readers online (Appendix A).

Ten out of the final 34 included trials did not provide numerical data for our planned meta-analytic assessments. However, mathematical findings presented here covered 93.7% (n = 10.097) of the total study population. Importantly, the calculation of pooled effect estimates for common efficacy parameters (e.g., Hgb and serum transferrin and ferritin levels) was not possible, even if six out of the analyzable 24 trials performed a stratified randomization of study participants on these parameters and other efficacy and non-efficacy variables (e.g., ESA therapy and transfusion requirements, study participants characteristics) [11,12,16,17,18,19,20]. The heterogenous definition of such variables was the main cause impeding this analysis.

All RCTs investigated were of moderate to high quality (Appendix A). Figure 2 shows pairwise meta-analysis of 19 trials, comparing exclusively IV and PO iron supplements for the combined outcome of non-response (i.e., Hgb increase of <0.5–1.0 g/dL, or initiation/intensification of ESA therapy, or increase/change of iron supplement, or requirements of blood transfusion). Figure 3 shows the Bayesian network diagrams built with all 24 trials, comparing different IV preparations to PO iron supplements. Overall, more preparations appear to have a protective effect against the combined outcome of non-response before the start of dialysis than once dialysis is initiated. Indeed, as depicted in Figure 4, 400 mg or more of iron sucrose per month (odds ratio (OR), 95% credible interval (CrI); 0.46, 0.30 to 0.68), 100 to 300 mg of iron sucrose per month (0.48, 0.31 to 0.77), 1020 mg of ferumoxytol per month (0.28, 0.16 to 0.47), and 750 to 1500 mg of ferric carboxymaltose per month (0.36, 0.24 to 0.53) were the most efficient formulations among CKD patients into the Kidney Disease—Improving Global Outcomes (KDIGO) glomerular filtration rate (GFR) categories G3A to G5, compared to PO iron supplements and no iron administration. Contrarily, only 400 mg or more of iron sucrose per month (0.13, 0.02 to 0.50) and 400 mg or more of iron dextran per month (0.08, 0.01 to 0.64) were efficacious among dialysis patients. At the pairwise level, heterogeneity was particularly evident (I² > 50%). Moreover, the asymmetry of funnel plots involving estimates on both patients in the KDIGO GFR categories G3A to G5 and dialysis patients was important (Egger’s test (t)/degrees of freedom (df)/p; −2.3591, 17, 0.0305).

Surprisingly, on the basis of the value of the surface under the cumulative ranking area (SUCRA), before starting dialysis, ferumoxytol (>0.9) and ferric carboxymaltose (0.808), respectively, were markedly different than iron sucrose preparations (<0.6) and the other iron supplements assessed (Table 1). In the group of chronic dialysis patients, such a difference between iron supplements was not perceived. The model chosen for calculating SUCRA values shows convergence, but a degree of inconsistency (Appendix A).

Finally, effect estimates presented here should be considered as provided by a low-quality body of evidence according to the Grades of Recommendation, Assessment, Development and Evaluation (GRADE) approach. Quality rating fell by two levels for heterogeneity and risk of reporting bias, even if there were not influence of indirectness in terms of participants/population, interventions, comparators and outcomes, nor of important imprecision in summary estimates (i.e., no wide confidence or credible intervals).

## 3. Discussion

An adequate body of evidence supports the efficacy of IV iron supplementation before and once dialysis is initiated, compared to PO iron supplementation. Nevertheless, in terms of treatment response, in CKD patients into the KDIGO GFR categories G3A to G5, different galenic forms of iron supplementation are efficacious, while some are more potent (e.g., ferumoxytol, ferric carboxymaltose). Clinical implications of these findings may question the use of less potent formulations, nevertheless, they may be used in patients needing to achieve looser goals (to decide in the clinical arena).

Our findings lead to an individualized treatment approach of IDA in CKD patients. Despite concerns regarding IV iron supplementation, such as anaphylaxis, bacterial infections, and atherosclerosis promotion [21], sustained Hgb level response observed in various systematic reviews and meta-analyses when compared to PO iron supplements justifies identification of the most performing ones in CKD patients [22,23,24,25,26]. This study was, thus, focused on assessing evidence on the efficacy of these nanomedicines by examining differences in treatment response between commercial IV iron supplements: pairwise and network meta-analyses were carried out to elucidate the individual effects of these drugs that conform the main intervention of iron supplementation.

In recent years, the introduction of new IV iron supplements have permitted the administration of larger doses of iron needed in CKD patients with IDA [27], constituting an important argument to use less PO iron preparations, even before dialysis [4]. Nevertheless, as of today, there is still no conclusive information on the efficacy of these new preparations in CKD patients: ferumoxytol [28] and ferric carboxymaltose [29,30] have an impact mostly on non-selected populations, according to three systematic reviews and meta-analyses, that may not be discerned from that of other iron supplements once glomerular filtration fall of 60 mL/min, as observed in another five evidence summaries studying CKD populations [22,23,24,25,26].

According to our findings, ferumoxytol and ferric carboxymaltose were associated with a better treatment response in terms of Hgb level increase and the absence of the need for other treatments of anemia in CKD patients. Such an impact is not mathematically comparable with that of other iron supplements. Our findings provide, thus, a solution to the need for enlarging the evidence body size concerning CKD, claimed by evidence summaries a few years ago [25,26,30], and for which this study may be considered an update.

This meta-analysis has been carried out according to a planned, registered, and prospectively updated systematic review protocol following current recommendations [31], as a clear signal of maintaining transparency in the systematic review process [32], avoiding future changes, which may be associated with reporting biases [33], and showing the suitability and non-duplicity of our analysis [34]. Nevertheless, various limitations should be mentioned. Publication bias is most likely the cause of the observed funnel plot asymmetry [35]. Publication and other reporting biases can lead to overly optimistic conclusions in a meta-analysis [36]. Heterogeneity should also be taken into account, as conclusions from a meta-analyses are less clear when the included studies have differing results [37]. Furthermore, summary estimates presented here included data from extension follow-up studies/post-hoc analyses of some of the eligible RCTs [6,7,8,9,10,11,12,13,14,15], which invite cautious interpretation, as findings from unplanned analyses are of lesser value [38]. Finally, this meta-analysis includes trials of less than 1000 participants, so our findings contribute to clarify false substantial effects reported by such small trials [39]: probably more research is needed to deny the absence of effects from 1500 mg or more of ferric carboxymaltose per month, which was observed in the 400 participants who underwent these doses, compared to the 3200 participants using lesser doses.

In conclusion, the new IV iron supplements ferumoxytol and ferric carboxymaltose appear to be the best performing preparations in CKD patients before dialysis. However, the other commercial iron supplements, such as iron sucrose or iron dextran, may continue to be used, especially in dialysis patients, and in all those in whom looser goals may be permitted. This study did not address safety concerns (e.g., anaphylactic reactions reported with ferumoxytol)—it was intended to measure efficacy of these nanomedicines on the basis of treatment response. Our analyses provide physicians with parameters for adequately following up patients requiring iron supplementation, and indirectly by making it possible to identify good responders among all treated with these medicines. Evidence-based treatment strategies may lead to individualized treatment strategies in CKD patients [40].

## 4. Materials and Methods 

This manuscript presents findings from pairwise and network meta-analyses carried out in accordance with the PRISMA recommendations [31], and meeting the PRISMA extension statement requirements for reporting of systematic reviews incorporating network meta-analyses of healthcare interventions [41]. For further details of our methods and their prospective character, our registered systematic review protocol (International prospective register of systematic reviews PROSPERO reference ID: CRD42020148155) is available for readers by clicking on the following link: https://www.crd.york.ac.uk/prospero/display_record.php?ID=CRD42020148155.

By using database-specific search strategies being developed with search terms related to participants/population, interventions, exposures, and the type of study to be included, MEDLINE via PubMed, Ovid and Web of Science, EMBASE via Elsevier’s Scopus, the Cochrane Controlled Register of Trials (CENTRAL), and other important databases of published studies via Web of Science, were searched up to September 2019. In addition, ClinicalTrials.gov, the EU Clinical Trials Register and the United Kingdoms’ ISRCTN registry, and relevant gray literature sources were searched. The reference lists of the included studies were scanned to identify all relevant studies cited by the included studies. The literature search was limited to the English language. Our full search strategy and search results are available for readers online (Panel S1).

As previously described [40], our strategy was focused on the identification of RCTs including their extension follow-up studies, and all varieties of post-hoc analysis that assessed individuals with normal kidney function (NKF)/CKD into KDIGO GFR categories G1 and G2, CKD patients into the KDIGO GFR categories G3A to G5, chronic dialysis patients, and kidney transplant patients. Nevertheless, in this study, our evaluations were centered on comparing evidence of the efficacy of IV and PO iron supplements: our main outcome was treatment response, which was defined by the increase in Hgb level and the need for other treatments of anemia in CKD (e.g., ESA therapy or blood transfusion).

After assessing risk of bias in the included studies [42], the overall OR with their 95% confidence interval (95% CI) for the outcomes of Hgb increase of ≥0.5–1.0 g/dL and the combined of non-response to iron supplementation (i.e., Hgb increase of <0.5–1.0 g/dL, or initiation/intensification of ESA therapy, or increase/change of iron supplement, or requirements of blood transfusion), were obtained (Mantel–Haenszel random-effect method), with evaluation of heterogeneity (χ², I²) and reporting bias risk (visual inspection of funnel plots and calculation of Egger’s test, if necessary) in summary estimates. Review Manager software (RevMan) version 5.3 (Cochrane Collaboration) and META-analysis package FOr R (METAFOR) version 2.4 (R project) were used for calculations at the pairwise level. Thereafter, OR with their corresponding 95% CrI for the combined outcome of non-response to iron supplementation were calculated via Bayesian network meta-analysis (Markov chain Monte Carlo simulation on vague priors random-effect method for ‘bad’ outcomes and zero values correction), with calculation of the value of SUCRA corresponding to each of the iron supplements described in studies eligible, and with verification of convergence (Brooks–Gelman–Rubin method) and inconsistency. NetMetaXL software (Canadian Agency for Drugs and Technologies in Health and Cornerstone Research Group) [43] was used for performing network meta-analysis. The analysis was performed for two subgroups: 1) patients in KDIGO GFR categories G3a to G5, and; 2) dialysis patients. Skewed and non-quantitative data was presented descriptively following the recommendations of the Centre for Reviews and Dissemination (University of York) [44]. Quality rating was performed by using GRADE [42]. 

## Figures and Tables

**Figure 1 pharmaceuticals-13-00085-f001:**
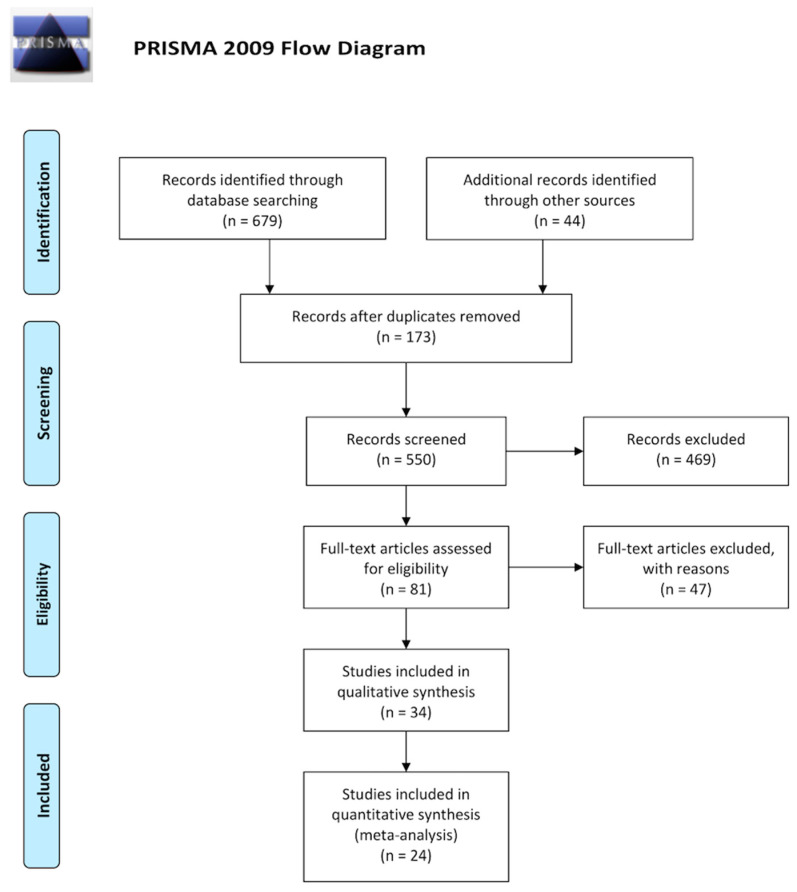
PRISMA flowcharts presenting our systematic review selection process for retrieving iron supplementation evidence on clinical trials. PRISMA, Preferred Reporting Items for Systematic Reviews and Meta-Analyses.

**Figure 2 pharmaceuticals-13-00085-f002:**
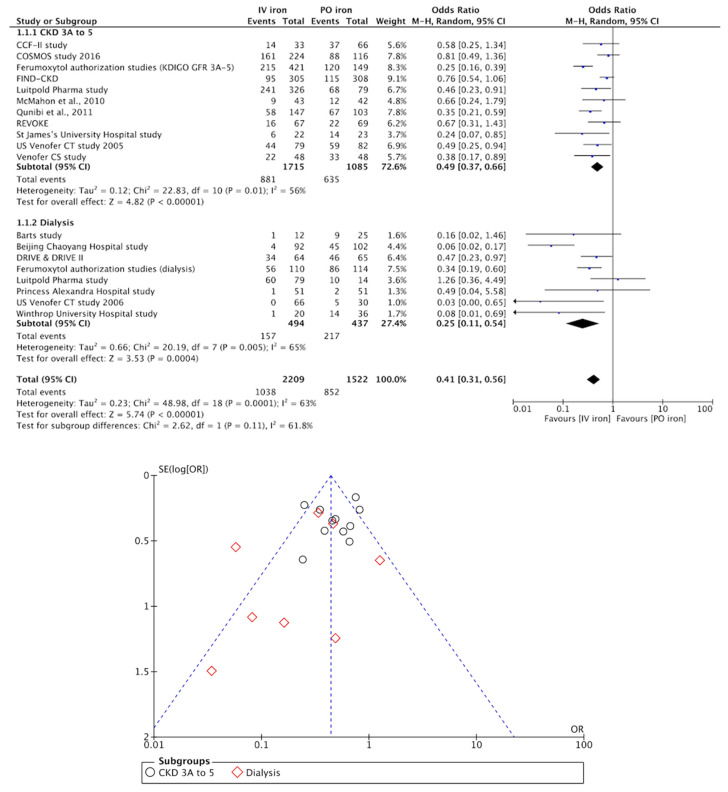
Forest and funnel plots showing effect estimates of IV versus PO iron supplements in the two subgroups conformed. CI, confidence interval; IV, intravenous; M-H, Mantel–Haenszel test; PBO, placebo; PO, per os; SE, standard error.

**Figure 3 pharmaceuticals-13-00085-f003:**
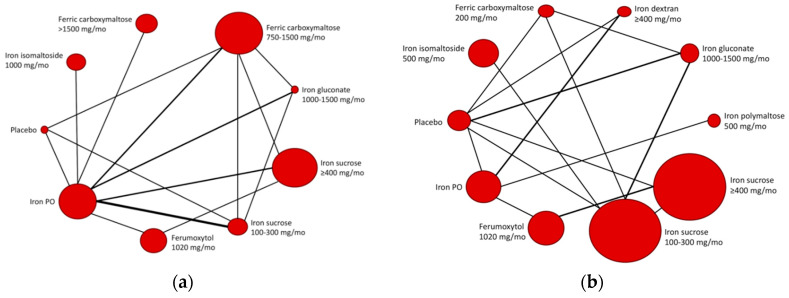
Bayesian network diagrams presenting the density in the comparisons (thickness of lines according to the number of RCTs in each comparison) and the competing iron supplements (node size according to the number of participants undergoing interventions) for the subgroups of (**a**) patients in the KDIGO GFR categories 3A to 5, and; (**b**) dialysis patients. GFR, Glomerular filtration rate; KDIGO, Kidney Disease—Improving Global Outcomes; RCT, randomized controlled trial.

**Figure 4 pharmaceuticals-13-00085-f004:**
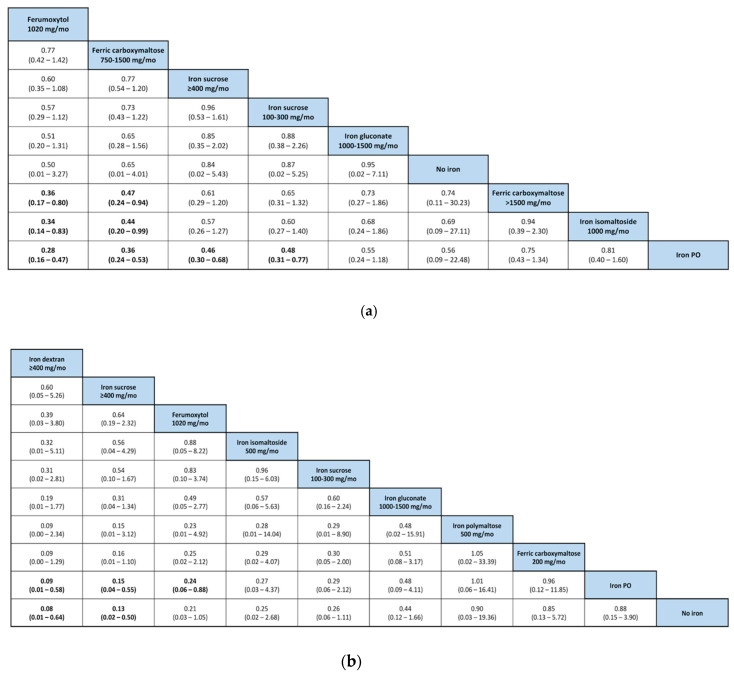
League tables showing ORs and 95% CrIs corresponding to the assessed IV iron supplements against the comparators in the subgroups of (**a**) patients in the KDIGO GFR categories 3A to 5, and; (**b**) dialysis patients. CrI, credible intervals; OR, odds ratio.

**Table 1 pharmaceuticals-13-00085-t001:** SUCRA-based ranking of iron supplements evaluated.

Iron Supplements ^†^	SUCRA ^‡^ CKD 3A-5/dialysis ^§^
Ferumoxytol 1020 mg/mo	0.926/0.673
Ferric carboxymaltose 750–1500 mg/mo	0.808/NA
Iron sucrose ≥400 mg/mo	0.598/0.840
Iron sucrose 100–300 mg/mo	0.567/0.614
Iron isomaltoside 500 mg/mo	NA/0.615
Iron gluconate 1000–1500 mg/mo	0.502/0.439
Iron polymaltose 500 mg/mo	NA/0.293
Ferric carboxymaltose >1500 mg/mo	0.280/NA
Iron isomaltoside 1000 mg/mo	0.248/NA
Iron P.O.	0.091/0.176

^§^ SUCRA values are expressed for each of the two subgroups conformed. ^†^ Iron supplements analyzed were ranked according to probabilities for being the best, the second best, the third best, and so on P(v=b), b=1,…,a following Markov chain Monte Carlo methods. ^‡^ SUCRA for each preparation v out of the *a* competing iron supplements requires calculation of the *a* vector of the cumulative probabilities $cum_{v,b}$ to be among the b best drug, b=1,…,a. Abbreviations: CKD, chronic kidney disease; NA, non-available; P.O., per os; SUCRA, surface under the cumulative ranking area.

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
