# Peer review of "The Impact of Iron Supplementation for Treating Anemia in Patients with Chronic Kidney Disease: Results from Pairwise and Network Meta-Analyses of Randomized Controlled Trials"

_pharmaceuticals, 2020, doi:10.3390/ph13050085_

Round 1

Reviewer 1 Report

This meta-analysis is showing the impact of IV vs PO and different form of iron supplementation among CKD patients.

The manuscript is well written and the message is clear. I have no further remarks about the results and methods as the authors discuss correctly the limits of this approach.

Minor comments :

Please improve the resolution of the tables and figures in the final version.

In Fig 2 : experimental = IV and control = PO ? could you specify Iron IV or Iron PO instead of experimental/control for sake of clarity.

Highlight in bold or in a different color the significant OR in Figure 4a/b

Author Response

Comments and Suggestions for Authors:

This meta-analysis is showing the impact of IV vs PO and different form of iron supplementation among CKD patients.

The manuscript is well written and the message is clear. I have no further remarks about the results and methods as the authors discuss correctly the limits of this approach.

Minor comments: 

Please improve the resolution of the tables and figures in the final version.

We very thank the reviewer for the comments. Visual quality of tables and figures with highlighting of critical findings and better explanations in figure and table legends have been Improved.

In Fig 2 : experimental = IV and control = PO ? could you specify Iron IV or Iron PO instead of experimental/control for sake of clarity.

The changes have been done to present more clearly the analysis of studies comparing exclusively IV and PO iron supplements depicted in Figure 2.

Highlight in bold or in a different color the significant OR in Figure 4a/b

The reviewer is right. Thank you. Done.

Reviewer 2 Report

Adler et.al.is a metanalysis looking at specifically the different responses of different IV iron

Very valuable and interseting findings about response to ferraheme being more potent in CKD 4/5 and iron dextran in ESRD.

The presentation is good, though English needs moderate improvement especially in introduction, the rest of paper has better diction and grammar.

Figure 3 can use a bit of clarification as to come to the conclusions presented in text.

The figures are beautifully rendered, and the science is sound.

The methods are clear.

The conclusions are valuable as far as clinical practice changing.

It was a pleasure to review this paper, congratulations on a good paper.

Author Response

Comments and Suggestions for Authors:

Adler et al. is a metanalysis looking at specifically the different responses of different IV iron.

Very valuable and interseting findings about response to ferraheme being more potent in CKD 4/5 and iron dextran in ESRD.

The presentation is good, though English needs moderate improvement especially in introduction, the rest of paper has better diction and grammar.

We very thank the reviewer for these comments and the compliment. Improvements have been made in the Introduction section and typos in other sections have been corrected. Please, note that in the Introduction section, now the background, the hypothesis and our study objective are presented more clearly and in different paragraphs.

Figure 3 can use a bit of clarification as to come to the conclusions presented in text.

Thank you very much. The reviewer is right. Figure foot have been entirely rephrased, and now explains network geometry for the two subgroups of the analysis in a more clear way, for better understanding of figure meaning.

The figures are beautifully rendered, and the science is sound. The methods are clear. The conclusions are valuable as far as clinical practice changing.

It was a pleasure to review this paper, congratulations on a good paper.

Once more, thank you for the comments and the compliment.